# The Influence of Lockdown on the Gambling Pattern of Swiss Casinos Players

**DOI:** 10.3390/ijerph18041973

**Published:** 2021-02-18

**Authors:** Suzanne Lischer, Angela Steffen, Jürg Schwarz, Jacqueline Mathys

**Affiliations:** 1School of Social Work, Lucerne University of Applied Sciences and Arts, 6002 Lucerne, Switzerland; Jacqueline.Mathys@hslu.ch; 2School of Business, Lucerne University of Applied Sciences and Arts, 6002 Lucerne, Switzerland; Angela.Steffen@hslu.ch (A.S.);

**Keywords:** COVID-19, gambling disorder, Swiss casino players, pandemic, playing behavior

## Abstract

The coronavirus pandemic (COVID-19) has had a major impact on most societies worldwide, including the closure of non-essential businesses in spring 2020. The present study considers its impact upon gambling behavior. Particularly, changes in self-reported gambling by Swiss, land-based casino players are examined. The main characteristics of respondents who played or did not play during lockdown are also investigated. This study is embedded in an ongoing longitudinal study that examines the gambling behavior of casino players at three points in time. All respondents who had participated in the first wave of the longitudinal study by the cut-off date (15th March 2020) were asked about their gambling behavior during lockdown in a supplementary online survey three weeks after the end of lockdown. A total of 55% of the 110 respondents reported having played during lockdown. Gambling intensity significantly decreased (*p* < 0.001) in our sample. Considering only those respondents (*n* = 66) who reported having gambled during lockdown, gambling intensity also decreased (*p* < 0.001), but online gambling significantly increased (*p* < 0.002). Those players who have increased their gambling activity require particular attention. It is important that casinos respond with appropriate player protection measures to those who have increased their gambling activity during the pandemic.

## 1. Introduction

The coronavirus-2019 (COVID-19) pandemic has had a major impact on most societies worldwide, including the closure of non-essential businesses in spring 2020. Likewise, gambling has been affected in many ways during the COVID-19 pandemic, including casino closures during the lockdown and the implementation of safety measures upon re-opening. Against this contextual background, it is generally assumed that the COVID-19 pandemic has had an impact upon gambling behavior [1,2,3,4]. Various influencing factors are discussed. One possibility is that the COVID-19 pandemic has led to dramatic changes to the amount of time spent at home, and likely increases in time spent online. Stay-at-home mandates and quarantines increase the consumption of digital forms of entertainment, especially online games and related activities (e.g., following sports broadcasts and video game streaming) [5,6]. Another plausible reason why more people have been gambling online could be that players who previously participated in land-based gambling migrated to online gambling during the casino closures [7,8]. Last but not least, there has been heavy promotion of a plethora of other gambling products including online casino games, online slot machines and virtual sports [9]. However, specific to pandemics, the same social distancing phenomena that may increase some types of gambling are likely to have opposite effect on other types: Pausing sport events means that there are fewer possibilities for betting, and closing terrestrial casinos, as well as restaurants and clubs with slot machines, obviously reduces these types of gambling opportunities [3,10].

Although evidence is still limited, an emerging discourse on the secondary impacts of COVID-19 has included the matter of gambling behavior. Thus, financial constraints and worries, social isolation and stress, which are likely to be particularly felt by people with insecure jobs or recent unemployment, may discourage entry into gambling or exacerbate problematic gambling [11]. While currently speculative, financial hardships may promote gambling, as individuals may be motivated to gamble to try to win money [2]. Moreover, many people gamble to relieve stress and boredom, which is especially important in times of a pandemic. However, this does not explain why some individuals then go on to gamble excessively, to the point where the behavior becomes problematic. It may well be that the gambling behavior increases stress, anxiety and depression, as the gambler has to bet increasingly larger amounts to cover losses or to receive the same level of emotional reinforcement from the activity [12]. Thus, whilst co-morbidities between gambling, depression and anxiety are well-evidenced, the direction of the effect is less clear. Depression can precede gambling, with gambling used to escape from or relieve negative emotions; however, the converse is also true—gambling can lead to financial and social difficulties that in turn lead to depression [13]. Issues related to the relationship between stress, anxiety and problem gambling are particularly relevant in the context of the COVID-19 pandemic. One study examining the emerging impact of COVID-19 on gambling during the first six weeks of emergency measures in Ontario highlighted that those fulfilling screening criteria for moderate or severe forms of anxiety and depression were more likely to have gambled online during the first six weeks of emergency measures and were classified as “high-risk gamblers” [4]. This finding is consistent with a previous study that noted comorbid relationships between mental health outcomes and problem gambling [14].

To date, there is only a limited body of scientific literature on the impact of lockdown upon gambling behavior. However, the potential for changes in land-based casino players’ gambling behavior during the pandemic (e.g., turning to alternative gambling products, increasing or reducing the intensity of gambling behavior) highlights the importance of observing players’ responses. In order to design adequate player protection measures, evidence in relation to this is indispensable. For this reason, the present study, involving a web-based survey dedicated to players of land-based casinos, was carried out three weeks after the end of the first lockdown in Switzerland. It explored the question of how the self-reported gambling behavior of land-based casino players changed during the lockdown due to the COVID-19 pandemic compared to the period before this. On the one hand, the study examined whether players turned to other forms of gambling. On the other hand, it examined whether a change in the intensity of gambling behavior can be observed. In addition, the sociodemographic characteristics, mental health characteristics and the gambling behavior measured by South Oaks Gambling Screen (SOGS) ratings of players who played during the lockdown were compared to those who did not. Special focus was given to the gambling behavior of players who were excluded from legal gambling.

## 2. Methods

### 2.1. Setting

Due to the continuing sharp rise in the number of coronavirus infections, the Swiss Federal Council declared an “extraordinary situation” on 16th March 2020. The associated ordinance placed massive restrictions on public life, as all non-essential businesses and services, including casinos, had to close with immediate effect. In compliance with safety precautions, casinos were allowed to restart their activities with limited services as of 6th June 2020. Moreover, all borders were closed until 15th June 2020 for entry into Switzerland by people with “non-essential purposes”. Because neighboring countries also kept their borders closed, there was no possibility for players to migrate into foreign casinos during the lockdown.

In Switzerland, gambling is regulated by the Federal Gambling Act, which entered into force in January 2019. With the revised law, licensed Swiss casinos have been given the opportunity to offer casino games online. The offering of lotteries and sports betting, both online and land-based, is reserved for two Swiss lottery companies [15]. In March 2020, five Swiss casinos were legally offering online gambling products. To keep unlicensed online gambling providers out of the market, web blocking was set up in July 2019. However, experience shows that it is easy for players to circumvent the blocking. In 2017, an estimated 8.6 percent of the Swiss population (over age 18) played casino table games on at least one occasion. In addition, 6.7 percent were reported to have played on slot machines (which are only permitted in casinos). Within the Swiss population, the annual problem gambling prevalence is estimated to be 2.8 percent for “high-risk gambling” and 0.2 percent for “pathological gambling” [16]. Since Switzerland carries out epidemiological studies on gambling only every five years, data for more recent years are not yet available. European past-year problem gambling prevalence rates varied between 0.12 percent and 3.4 percent [17]. Previous studies indicated that the prevalence of problem gambling in Switzerland is comparable to that in other European countries [18]. However, given that epidemiological studies are based on different screening instruments, it is difficult to draw firm conclusions from such comparisons [17].

The Federal Gambling Act requires every casino to develop a clear prevention strategy. More precisely, casino staff must follow guidelines and use checklists to identify at-risk players and engage them in a dialogue about their gambling behaviors. This approach aims to help casinos employees implement appropriate intervention measures. Bans are imposed if proof can be found or there is strong suspicion that, due to their gambling behavior, specific players are maintaining excessive debts, placing bets that are disproportionate to their financial circumstances, or experiencing other disruptions. On the other hand, players can also ask to be self-excluded. The ban applies throughout Switzerland, and the casino must disclose players’ identities to all other Swiss casinos [19]. It is an overarching exclusion system, i.e., a banned player is excluded from land-based gambling and online gambling as well as online lotteries and sports betting [15].

### 2.2. Study Procedures

The present study, hereinafter referred to as the “COVID-19-study”, is embedded within an ongoing online longitudinal study, which evaluates the influence of voluntary and imposed exclusion as a player protection measure. The research is currently being conducted in all three linguistic regions of Switzerland: German, French, and Italian. Within the longitudinal study, players who are banned from Swiss Casinos complete an online questionnaire on three occasions, at six month intervals. The excluded players are recruited by flyers that are handed out by the casino staff to raise awareness of the project. Players who have agreed to participate subscribe themselves to a website. They then receive an e-mail containing a link that gives them access to the online survey. Six months, and then twelve months later, they receive e-mails for the second and third surveys, respectively. Participants receive a shopping voucher with the value of CHF 20 for each survey they complete. The first survey (T1) takes place at the time of exclusion. The responses of excluded players are compared to a control group of non-banned casino players who complete the questionnaire at the same time intervals. The non-banned players are also recruited through flyers distributed by casino employees. The players’ data are stored on a secured server at the Lucerne University of Applied Sciences and Arts. The first wave of surveys (T1) was launched in autumn 2019 [20]. For the “COVID-19-study”, a total of 171 respondents (excluded players and controls) who had participated in the first survey wave (T1) by the cut-off date of 15th March 2020 were invited by email to provide feedback on their gambling behavior during lockdown. The invitation was sent on 29th June 2020, three weeks after the reopening of land-based casinos. This procedure allowed us to investigate the behavior of casino players before (T1) and during the lockdown. Participation in the “COVID-19-study” was also rewarded with a shopping voucher worth CHF 20.

### 2.3. Participants

The invitation to take part in the “COVID-19-survey” was sent to 90 German-speaking, 48 French-speaking and 33 Italian-speaking land-based casino players in Switzerland. The response rate was *n* = 110 (64.3 percent). As mentioned above, the entire sample had already participated in the first wave of the longitudinal study (T1) and were identified as trustworthy, in the sense they have not tried to participate multiple times and had not taken an unrealistically short time to complete the survey. The answers were matched with information from the first longitudinal survey wave (T1), using participants’ email addresses as identifiers.

### 2.4. Measures

**Sociodemographic data.** The longitudinal survey consisted of demographic questions related to age, gender, marital status, and the highest level of education attained. Respondents were also asked about their employment status and net income.

**Gambling behavior.** To measure gambling behavior, the questionnaire contained questions on respondents’ participation in the different types of gambling products available in Switzerland during the past six months. Since the Swiss gambling market differs from foreign markets, the questionnaire items for the study had to be constructed. The items were developed using a multi-stage Delphi procedure in which experts were interviewed. The questions were tested in a pretest with players from several land-based casinos.

Study participants were asked to specify their gambling frequency and gambling duration for each of the 26 game types specified. The different forms of gambling were casino games, lotteries, and sports betting in Switzerland (both land-based and online). Furthermore, questions were asked about land-based gambling offers from abroad (casinos, arcades, lotteries, and sports betting) as well as online gambling offers from international operators. The survey also asked about gambling in private settings or in so-called “backrooms”. Gambling frequency was investigated using six categories (less than once a month, one to three times per month, one to two times per week, three to four times per week, five to six times per week, daily). Gambling duration included six categories (less than one hour, 1 to 2 h, 3 to 4 h, 5 to 6 h, 7 to 8 h, more than 8 h). Since the study evaluated gambling exclusion as a measure of player protection, participants were asked whether they were excluded and if so, questions were posed about the exclusion process. In addition to these self-constructed questions, the questionnaire contained the following validated batteries.

**Motivation for Gambling.** The players were also asked about their motivations for participating in the game. Eleven items were included, nine of which were taken from the so-called “catamnesis study” [21]. The final two items were self-constructed by the authors. In place of the original version with multiple choice selections, the items were rated on a five-point Likert scale from “does not apply at all” to “fully applies”. The items in the Italian and French questionnaires were translated according to guidelines from the European Social Survey Program for the translation of questionnaires [22].

**Problem Gambling.** In addition to the questions above, the level of potential gambling problems was measured using the South Oaks Gambling Screen (SOGS), where each of the statements concerns the past six-month period [23]. Responses to the 20 items are summed, and endorsement of five or more items is taken to indicate the presence of pathological gambling [24]. Authorized German, French and Italian versions were used for the survey [25,26,27]. Reliability tests for the SOGS scale revealed high internal consistency (Cronbach’s alpha = 0.82). The SOGS was administered in the first wave of the survey (T1) in order to categorize participants’ gambling behavior.

**Mental Health.** Mental health concerns were investigated using the four-item Patient Health Questionnaire (PHQ-4). The PHQ-4 is a rapid self-reported measure to investigate depression and anxiety. The PHQ-4 consists of two subscales, each containing two items for depression and anxiety with scores ranging from 0 to 6 points for each subscale [28]. The PHQ-4 consists of the first two items of the Generalized Anxiety Disorder scale (GAD-7) [29] and the first two items of the Patient Health Questionnaire (PHQ-9) [30]. Respondents rate their symptoms using a four-item Likert rating scale ranging from 0 (not at all) to 3 (nearly every day), and the total score ranges from 0 to 12. The severity of clinically relevant depression and/or anxiety according to the PHQ-4 score is to be interpreted as follows: normal (0–2), mild (3–5), moderate (6–8), severe (9–12) [31]. The PHQ-4 is a well-validated screening instrument, and it demonstrated a high internal consistency (Cronbach’s α = 0.82) [28]. For the German, Italian and the French versions of PHQ-4 the instruments PHQ-9 and GAD-7 were taken from Pfizer [32]. This site includes liability disclaimers, the PHQ and Generalized Anxiety Disorder screening tests (in multiple languages), as well as the instruction manual and relevant bibliography.

**Life Satisfaction.** Respondents’ global life satisfaction was investigated with the Life Satisfaction-1 short scale (L-1). This is a modified version of the Socio-Economic Panel (SOEP), which measures general life satisfaction, using only one item. The answer format of the L-1 consists of a unipolar, 11-point scale ranging from “not at all satisfied” (0) to “completely satisfied”. The reliability of L-1 has been estimated using the retest method and has been reported as *r* = 0.67, with an average repeat interval of six weeks [33]. The Italian and French translations of L-1 were taken from the Swiss lives’ cohort panel W3 [34].

**Gambling during COVID-19.** To investigate the influence of the COVID-19 pandemic on gambling behavior, questions on gambling behavior (game type, frequency, and duration) were asked again. As certain game types were limited during lockdown (e.g., land-based casinos, offers from abroad) only 18 different game types were included in the survey. Frequency was investigated using six categories (less than once per month, one to three times per month, one to two times per week, three to four times per week, five to six times per week, daily). Gambling duration included six categories (less than one hour, 1 to 2 h, 3 to 4 h, 5 to 6 h, 7 to 8 h, more than 8 h). The other data (socio-demographic data, self-reported problem gambling, mental health, life satisfaction, motivations for gambling and whether participants were currently excluded), were taken from the longitudinal study (T1).

**Gambling intensity index.** As mentioned above, participants’ gambling frequency and gambling duration were asked about for 18 different game types. To measure changes in playing behavior between the times before and during the lockdown, the following approach was used: Frequency scores reported for each game type were summed to create a total frequency score. Likewise, total duration scores were calculated by adding the duration of play reported for each game type. Furthermore, in order to consider both gambling frequency and duration, in combination, a self-constructed instrument was applied (gambling intensity index). The gambling intensity index was calculated by first multiplying a participant’s frequency and duration scores for each game they had played. These values were then added over all 18 games. The resulting index ranged from 0 to 128 in the present sample.

### 2.5. Statistical Methods

The main results were derived from two types of analyses: A time comparison of respondents’ gambling behavior before and during lockdown, and a comparison of reported characteristics for respondents who gambled during the lockdown to reported characteristics of those who did not. The number of respondents who gambled during lockdown and the gambling intensity indexes for different types of online and offline gambling served as the main dependent variables. Changes in respondents’ gambling participation and gambling intensity during the lockdown were analyzed using McNemar’s chi-square and Wilcoxon signed-rank tests for different types of gambling. The dependent variables were further analyzed in terms of potential risk factors, using chi-squared bivariate analyses for categorical variables and Wilcoxon–Mann–Whitney tests for continuous variables. To investigate the impact of potential risk factors (respondents’ demographic characteristics, potential gambling problems (SOGS, gambling exclusion) and wellbeing indicators (L-1, PHQ-4) on the dependent variables, chi-squared bivariate analyses were used for categorical variables and Wilcoxon–Mann–Whitney tests were used for continuous variables. Further bivariate analyses investigated potential differences between respondents who gambled or did not gamble during the lockdown regarding their previous gambling behavior, gambling motives, and mental health. All calculations were carried out in Stata 16.

### 2.6. Ethics

The Swiss Ethical Authority decided that the project did not require formal ethical approval, since it does not involve research on human diseases or the structure and function of the human organism (file number Req-2019-00060). The data management plan was approved by the Swiss National Science Foundation.

## 3. Results

### 3.1. Sample Characteristics

A total of *n* = 110 participants took part in the Covid-19 study. Table 1 shows the main characteristics of the sample. Across the three Swiss linguistic regions, 85 males and 25 females responded to the survey. The distribution of the SOGS scores (mean = 2.2) reveals that 36.4 percent of respondents were classified as having no problems with gambling, 41.8 percent (*n* = 46) showed somewhat problematic gambling behavior and 19.1 percent (*n* = 21) were classified as probable pathological players, at baseline (T1). Seventeen respondents were excluded from gambling. The excluded players showed significant higher SOGS scores (mean = 5.7) than non-excluded players (mean = 1.5, *p* < 0.001); the mean L-1 life satisfaction score was 6.6 out of a maximum of 10 points. According to the distribution of the PHQ-4 score, 23.6 percent of the respondents showed moderate and 14.6 percent showed severe signs of a clinically relevant depression and/or anxiety before lockdown (mean = 4.4 out of 12).

### 3.2. Altered Playing Behaviour during the Lockdown

Fifty-five percent of the participants (*n* = 61) were gambling during lockdown. Table 2 illustrates the number of respondents who participated in different types of gambling (land-based and online) before and during the lockdown. Although casinos were closed and gambling abroad was not possible, there were several land-based gambling opportunities in addition to online gambling. These included lotteries, games in private settings and unlicensed gambling in so-called “back rooms” and Tactilos. These are electronic lottery devices, which are available in bars in the French-speaking part of Switzerland. They have been available since the reopening of bars in May 2020. Nonetheless, it is not surprising that during the lockdown, there was a significant decrease in the number of land-based gambling activities. At the same time, the McNemar’s chi-square statistic suggests no increase in online gambling during the lockdown compared to the time before the lockdown. According to the results of the survey, land-based casino players did not migrate to online gaming services.

To further investigate the gambling intensity, i.e., the frequency and duration of respondents’ gambling behavior, the gambling intensity before and during lockdown was calculated. The gambling intensity refers to the sum product of a person’s gambling frequency and duration across all types of gambling (see Section “Instruments”). Table 3 shows respondents’ average gambling intensity online and offline, before and during the lockdown.

As suggested by the prior analyses, the gambling intensity significantly decreased. The Wilcoxon signed-rank test indicated that only 15 respondents reported a higher gambling intensity during lockdown. Despite the significant decrease in land-based gambling in Switzerland and abroad, overall respondents did not gamble more excessively online during the lockdown. However, when considering only those respondents who reported having gambled during the lockdown (*n* = 61), online gambling significantly increased. Additional analyses showed that this includes both a significant increase in online gambling frequency (*p* = 0.003, Wilcoxon signed-rank test) and duration (*p* = 0.008, Wilcoxon signed-rank test).

Figure 1 provides a more detailed analysis of respondents’ participation in different types of gambling before and during the lockdown. Not surprisingly, McNemar’s chi-square statistics reported significant changes for the relative number of respondents gambling in casinos (*p* < 0.001) and gambling abroad (*p* < 0.001) during lockdown. However, the participation in all other types of gambling did not change significantly. Thus, the pandemic has not interrupted gambling, but has merely changed how it is taking place.

### 3.3. Playing Behaviour during Lockdown and Personal Chracteristics

To identify potential risk factors for gambling during the emergency measures, Table 4 shows the personal characteristics of respondents who did or did not participate in any form of gambling during the lockdown.

There are no significant differences between respondents who gambled during the lockdown and those who did not, with regard to gender, age, or linguistic region. However, individuals who gambled during the lockdown were significantly more often in a relationship (45.9% vs. 12.2%) and had higher net incomes (29.5% vs. 8.2% earning more than 9000 Swiss francs per month; equivalent to 8230 Euros) than individuals who did not participate in any gambling (effect sizes: Cohen’s d = 0.41 for family status; Cohen’s d = 0.71 for income levels). There is also a significant difference in the gambling behavior of excluded compared to non-excluded respondents, with excluded players being significantly less likely to participate in any type of gambling during the lockdown (*p* = 0.019, Cohen’s d = 0.46).

Gambling participation during the lockdown did not correlate with potential gambling problems, captured by the SOGS score. In fact, the mean SOGS score was lower for respondents who were gambling during the lockdown (2.03 vs. 2.36, *p* = 0.874). This is due to the fact that the excluded players, who had higher SOGS values, played less during lockdown. Without considering excluded players in the analysis, the higher SOGS scores for respondents who gambled during the lockdown were marginally significant (*p* = 0.082). In contrast to the SOGS scores, the results indicate significant relationships between gambling participation and respondents’ mental health. Players who gambled during the lockdown reported significantly stronger symptoms of depression and anxiety (mean PHQ-4 = 5.83 vs. 2.63, *p* < 0.001, Cohen’s d = 1.06) and significantly lower life satisfaction (mean L-1 = 6.18 vs. 7.12, *p* = 0.002, Cohen’s d = 0.06) at the time of T1.

### 3.4. Intensity of Playing Behaviour during Lockdown

The following analyses in Table 5 explore a potential relationship between gambling behavior before the lockdown and during lockdown. Not surprisingly, respondents who were gambling during lockdown showed higher gambling intensity scores at T1 (i.e., the sum product of a person’s gambling frequency and duration across all types of gambling) overall, and for different types of gambling (land-based in Switzerland, abroad, and online). The differences were also significant with regard to gambling frequency and gambling duration. Hence, players who were gambling during the lockdown tended to be more intensive players in general.

### 3.5. Motives of the Players Who Played during Lockdown

While the self-reported primary motives did not differ between respondents who gambled during the lockdown and those who did not, social and psychological motivations for gambling, such as being with friends or distraction from boredom and stress, played a greater role for respondents who gambled during the lockdown. Wilcoxon–Mann–Whitney tests confirmed significant differences for seven out of 11 gambling motives, as indicated by the error bars in Figure 2. 

## 4. Discussion

The pandemic has not interrupted gambling but has merely changed how it is happening. Fifty-five percent of the questioned land-based casino players continued to take part in gambling during the lockdown. Not surprisingly, these players have shown more intensive gambling behavior overall. Furthermore, they were significantly more likely to participate in online gambling. Without taking the excluded players into account in the analysis, the SOGS scores of the respondents who played during the lockdown were also significantly higher, albeit only marginally. Furthermore, the players who gambled during the lockdown showed significantly stronger symptoms of depression and anxiety. Additionally, with regard to the motivations for gambling, there were differences: social and psychological aspects, such as being with friends or distraction from boredom and stress, played a greater role for respondents who gambled during the lockdown. This result is in line with previous research showing that increased gambling remains associated with higher problem gambling severity [1]. Players who increased their gambling therefore merit special attention. Existing literature did show that both anxiety and boredom are associated with problem gambling, both of which can be expected to increase during uncertain and threatening times, and in connection with social distancing procedures introduced to curb disease transmission [3]. However, regarding the gambling behavior of excluded players, the findings of the present study differ from previous research. According to a Swedish population study, four percent of respondents increased their gambling during the first wave of COVID-19. Within the group, a surprisingly high proportion (28%) had a history of self-exclusion [1]. In contrast to the Swedish study, gambling participation by the excluded players decreased during lockdown. The most plausible explanation is that gambling offers to which banned players can divert were not accessible during the lockdown. A previously conducted study suggests that excluded players often switch to gambling offers in neighboring countries [19]. However, due to the small sample of excluded players (*n* = 17), the results should be interpreted with caution.

However, according to the present sample, general participation in gambling decreased during the lockdown. Moreover, the trend in which players participating in land-based gambling migrated to online gambling due to the limited playing possibilities during lockdown cannot be confirmed. These results are in line with previous research suggesting no major transition effects [1,3]. One possible explanation could be that the “typical” casino player only has a limited interest in online gambling. In addition, there are also various alternative options in land-based gambling, even during the lockdown. For example, the respondent players stated that they had participated in land-based lotteries including Tactilos as well as gambling in a private setting. However, as mentioned above, considering only those respondents who reported that they gambled during the lockdown (*n* = 61), online gambling significantly increased. Overall, online gambling in Switzerland requires special attention. Licensed online gambling has only been permitted in Switzerland since July 2019. The amplified volume of advertising as a result of the opening of the market in autumn 2019 was closely followed by the COVID-19 crisis. There is a strong possibility that many players who had not previously played or participated in online gambling before the pandemic have been attracted. Thus, industry reactions to new players or repeat players who are gambling more intensively merit careful monitoring [11].

## 5. Strengths and Limitations

Studies into gambling behavior before, during and after the COVID-19 pandemic are needed to capture changing behaviors and the changing influence of risk factors. Considering that the COVID-19 pandemic is a long-term problem, it is important that evidence of problematic gambling is gathered, as effective preventive measures can only be developed on the strength of this information. Whilst providing some useful insights, the present study has several limitations. The sample of 110 players is rather small and as a result the study did not consider certain variables, which are likely to present themselves as invalid when using a small sample size. Furthermore, the respondents reported receiving a relatively high income, which raises questions about the representativeness of the sample. Furthermore, the groups of excluded and non-excluded players are very disproportionate in size. Since the group of excluded players is considered particularly vulnerable, this unequal distribution, which only allows non-parametric tests, is also an important limitation. In order to assess the intensity of gambling, a gambling intensity index was developed for this study. However, the question of whether the newly developed instrument is suitable in terms of validity and what psychometric characteristics it exhibits needs to be clarified in subsequent research. Another important limitation is that the respondents were mostly people who had originally participated in land-based casino games. Therefore, no conclusions can be reached about players who originally participated in online casino games or online sports betting. Data on gambling behavior of these sub-populations would be valuable, especially in times of a pandemic. Finally, yet importantly, it should be emphasized that self-reporting of behaviors is also prone to a degree of response bias, which is a typical limitation of survey methods and designs.

## 6. Conclusions

The present article refers above all to land-based casino players. Overall, their gambling activity has decreased. However, those players who have increased their gambling activity require special attention. In Switzerland, as elsewhere, one of the objectives of gambling legislation is to avoid economic and social harm to players. Within a context of great economic and social uncertainties, this objective takes on an even more important meaning. The COVID-19 pandemic has negative financial consequences for a considerable proportion of the working population. Against this background, the early detection of players who are gambling beyond their financial means becomes even more crucial. Players with risky gambling behavior must be observed and addressed by casino staff. The measure of exclusion from the game must be applied to players who have debts or cannot afford to continue playing. However, in the context of early detection, the focus should not be exclusively on financial aspects. Players who intensify their gambling behavior or who are developing addictive behavior must be addressed and made aware of professional addiction support services.

The often-expressed concern that land-based players would migrate to online gaming during the lockdown due to limited gaming opportunities could not be confirmed by this sample. However, the growing online gambling market requires special attention, especially in times of crisis. Moreover, the influence of advertising on the gambling behavior of vulnerable players should be investigated. Future studies should be carried out in order to closely observe potential changes in gambling behavior and the associated risk factors during the ongoing course of the current pandemic.

## Figures and Tables

**Figure 1 ijerph-18-01973-f001:**
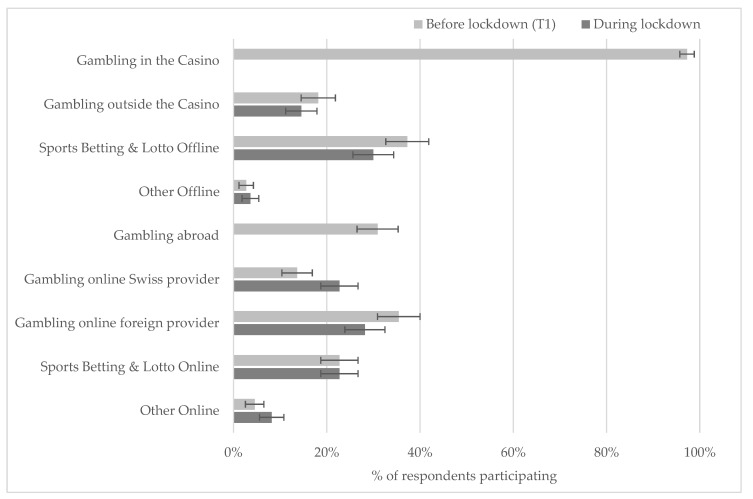
Relative Participation in Different Games Before and During Lockdown (*n* = 110). Note: The bars indicate the proportion of respondents who participated in each type of gambling. The error bars represent the mean +/− the standard error of the mean.

**Figure 2 ijerph-18-01973-f002:**
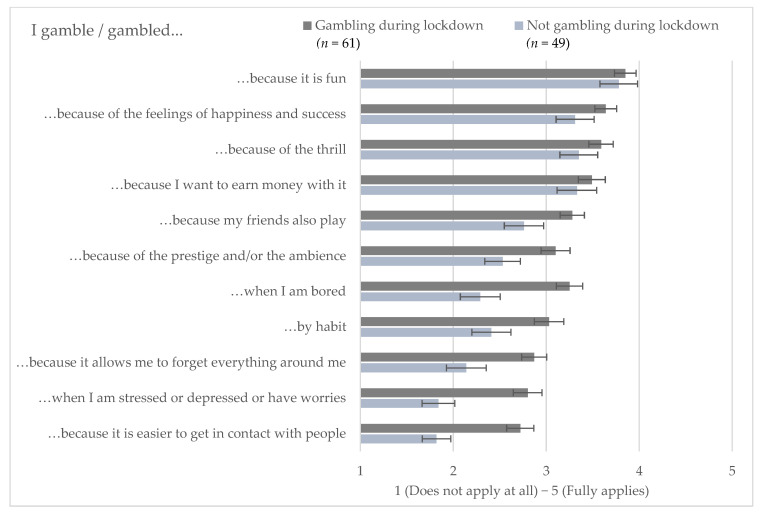
Gambling motives for respondents who gambled or did not gamble during lockdown. The bars indicate the mean values of five-point Likert scales. The error bars represent the mean +/− the standard error of the mean.

**Table 1 ijerph-18-01973-t001:** Sample Characteristics Measured at T1 (*n* = 110).

**Sample Characteristics**	***n* (%)/Mean (SD)**
**Gender**	
Female	25 (22.7%)
Male	85 (77.3%)
Age, mean (SD)	33.50 (11.64)
**Linguistic Region**	
German	64 (58.2%)
French	30 (27.3%)
Italian	16 (14.5%)
**Marital Status**	
Single	36 (32.7%)
In a relationship	34 (30.9%)
Married/In a registered partnership	17 (15.5%)
Divorced/Dissolved registered partnership	1 (1%)
No response	22 (20%)
**Education**	
Compulsory schooling	15 (13.6%)
Apprenticeship	46 (41.8%)
Diploma/College	26 (23.6%)
University degree	16 (14.6%)
Other	7 (6.4%)
**Net Income** (**Swiss Francs, per Month**)	
Less than 3000	27 (24.6%)
3000–7000	30 (27.3%)
7001–9000	26 (23.6%)
More than 9000	22 (20%)
No response	5 (4.6%)
**Gambling Exclusion**	
No exclusion	93 (84.5%)
Exclusion	17 (15.5%)
L-1, mean (SD)	6.6 (1.63)
**SOGS**, mean (SD)	2.19 (2.64)
**SOGS Categories**	
No problem	40 (36.4%)
Somewhat problematic	46 (41.8%)
Probable pathological gambling	21 (19.1%)
No response	3 (2.7%)
PHQ-4, mean (SD)	4.4 (3.39)
**PHQ-4 Categories**	
Normal	43 (39.1%)
Mild	25 (22.7%)
Moderate	26 (23.6%)
Severe	16 (14.6%)

Note: The table reports the absolute and relative number of participants per category for categorical variables, and the mean values and standard deviations for continuous variables; SOGS = South Oaks Gambling Screen; PHQ-4 = Patient Health Questionnaire.

**Table 2 ijerph-18-01973-t002:** Participation in Online and Offline Gambling Before and During Lockdown (*n* = 110).

Game Types	Before Lockdown (T1)	During Lockdown	Difference before–during	*p*-Value
**Land-based gambling Switzerland**	109 −99.10%	43 −39.10%	66 (60% points)	<0.001
**Land-based gambling abroad**	34 −30.90%	0 0.00%	34 (30.9% points)	<0.001
**Online gambling**	63 −57.30%	59 −53.60%	4 (3.7% points)	0.571

Note: The table reports the absolute and relative frequencies of gambling participation before and during the lockdown and the difference between these. The *p*-values in the last column show the chi-square statistics of the McNemar tests.

**Table 3 ijerph-18-01973-t003:** Gambling intensity for different categories of gamblers before and during the lockdown.

**Games Types**	**All Respondents (*n* = 110)**			**Respondents Gambling** **during the Lockdown (*n* = 61)**		
	**Before Lockdown**	**During Lockdown**	***p*-Value**	**Before Lockdown**	**During Lockdown**	***p*-Value**
	**Mean (SD)**	**Mean (SD)**		**Mean (SD)**	**Mean (SD)**	
**All Games**	16.64	5.92	<0.001	17.91	10.83	<0.001
	−16.36	−7.92		−11.3	−7.85	
**Land-based Gambling Switzerland**	11.8	1.97	<0.001	12.41	3.58	<0.001
	−11.28	−3.51		−6.53	−4.09	
**Land-based Gambling Abroad**	1.1	0	<0.001	0.57	0	
	−2.93	0		−1.42	0	<0.001
**Online Gambling**	3.77	3.94	00.241	4.8	7.2	0.002
	−7.24	−5.81		−7.81	−6.19	

Note: The table reports the mean gambling intensity index before and during the lockdown and the significance of the difference between these. The *p*-values in the last column were calculated using a Wilcoxon signed-rank tests.

**Table 4 ijerph-18-01973-t004:** Comparison between respondents who gambled and did not gamble during lockdown.

**Sample Characteristics**	**Not Gambling during Lockdown**	**Gambling during Lockdown**	***p*-Value**
	**(*n* = 49)**	**(*n* = 61)**	
**Gender**			0.19
Female	14 (28.6%)	11 (18.0%)	
Male	35 (71.4%)	50 (82.0%)	
**Age**, mean (SD)	33.25 (13.06)	33.69 (10.49)	0.56
**Language Region**			0.348
German	31 (63.3%)	33 (54.1%)	
French	10 (20.4%)	20 (32.8%)	
Italian	8 (16.3%)	8 (13.1%)	
**Marital Status**			0.002
Single	19 (38.8%)	17 (27.9%)	
In a relationship	6 (12.2%)	28 (45.9%)	
Married/In a registered partnership	8 (16.3%)	9 (14.8%)	
Divorced/Dissolved registered partnership	1 (2.0%)	0 (0.0%)	
No response	15 (30.6%)	7 (11.5%)	
**Education**			0.676
Compulsory schooling	6 (12.2%)	9 (14.8%)	
Apprenticeship	22 (44.9%)	24 (39.3%)	
Diploma/College	10 (20.4%)	16 (26.2%)	
University degree	9 (18.4%)	7 (11.5%)	
Other	2 (4.1%)	5 (8.2%)	
**Net income** (**Swiss Francs, per Month**)			<0.001
Less than 3000	17 (34.7%)	10 (16.4%)	
3000–7000	21 (42.9%)	9 (14.8%)	
7001–9000	4 (8.2%)	22 (36.1%)	
More than 9000	4 (8.2%)	18 (29.5%)	
No response	3 (6.1%)	2 (3.3%)	
**Gambling Exclusion**			0.019
No exclusion	37 (75.5%)	56 (91.8%)	
Exclusion	12 (24.5%)	5 (8.2%)	
**SOGS,** mean (SD)	2.36 (2.99)	2.05 (2.35)	0.874
**L-1**, mean (SD)	7.12 (1.73)	6.18 (1.42)	0.002
**PHQ-4**, mean (SD)	2.63 (2.58)	5.83 (3.30)	<0.000

Note: The table shows the number of participants who gambled or did not gamble during the lockdown, per category for each of the categorical variables. For continuous variables the mean and standard deviation are reported. The second column reports the predicted marginal effects of the variables on the probability that an individual had gambled during lockdown. The last column shows the p-values from χ2 tests for categorical variables or Wilcoxon–Mann–Whitney tests for continuous variables.

**Table 5 ijerph-18-01973-t005:** Gambling participation during lockdown and previous gambling intensity.

**Gambling Intensity**	**Gambling during** **Lockdown**	**Not Gambling during Lockdown**	***p*-Value**
	**(*n* = 61)**	**(*n* = 49)**	
**Gambling Intensity Index** (**T1**)	17.91 (11.30)	15.04 (21.12)	<0.001
**Gambling Frequency Index** (**T1**)	7.12 (4.16)	6.50 (5.76)	0.024
**Gambling Duration Index** (**T1**)	8.00 (4.05)	7.43 (6.66)	0.045
**Gambling Intensity Index Land-based Gambling Switzerland** (**T1**)	12.41 (6.53)	11.04 (15.35)	<0.001
**Gambling Intensity Index** **Land-based Gambling Abroad** (**T1**)	0.57 (1.42)	1.77 (4.05)	0.010
**Gambling Intensity Index** **Online Gambling** (**T1**)	4.80 (7.81)	2.45 (6.29)	<0.001

Note: The table reports means and standard deviation in parentheses. The *p*-values were obtained from Wilcoxon–Mann–Whitney tests.

## Data Availability

The data will be published at https://forsbase.unil.ch/ as soon as the longitudinal study is finalised.

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
