# Peer review of "The Influence of Lockdown on the Gambling Pattern of Swiss Casinos Players"

_ijerph, 2021, doi:10.3390/ijerph18041973_

Round 1

Reviewer 1 Report

Thanks for asking me to review this interesting manuscript. It's one of the few studies that has explored gambling in relation to COVID-19. In general, behavioural/process addictions receive less research interest than substance use disorders, so it is wonderful to see such a piece. As it stand, I do not have any methodological concerns with the study except for the small sample size; however, it was not a relevant limitation as there was statistical significance in most of the comparisons.

Author Response

Thank you for the appreciative review.

Reviewer 2 Report

The article discusses a very interesting and up-to-date topic. Nevertheless, it has several shortcomings.

The text gives the impression of a research report rather than a scientific study. Among other things, it lacks clearly formulated objectives and does not indicate which gap the researchers are to fill. The authors did not formulate research questions or research assumptions. I realize that the bibliography on pandemic gambling is limited at this point in time, but it would be useful to expand the rather meager literature discussed, including more extensive coverage of, for example, the impact of difficult/stressful situations on compulsive behavior and addictions.

It is a pity that the results concerning mental health, life satisfaction, motivations for gambling was taken only from the longitudinal study (T1), i.e. the time before lockdown. Examining these factors again could have been an interesting way to show if there were changes in the subjects at the time of lockdown. Then more could have been said about the determinants of changes in gambling patterns during the epidemic. Especially since the motivation for online gambling may be different from land-based gambling.

The practical implications are rather general and wishful thinking, e.g. "It is important that casinos respond with appropriate player protection measures to those players who increase their gambling activity in times of crisis. ”. The question arises what protection measures on the part of casinos would be. It seems advisable to elaborate on the practical implications.

There are several editorial errors e.g:
5 Strengths and limitations
6 Conclusion Strengths and limitations

Author Response

Thank you for the appreciative review. Subsequently you find our response to the comments you have raised.

Objectives and research gap
The principal research interest has been elaborated and the research questions have been specified. Furthermore, we explain where we have identified research gaps and how we intend to close them.

Literature
The state of research was expanded, and additional aspects were included, such as the explanations on the association between stress and problem gambling.

Representation of mental health issues over time
As you mention, the results of the change would be very interesting, and in fact we had calculated them. However, there are serious concerns of a methodological nature that led us not to publish the results.
Although we collected mental health and life satisfaction at both time points, we cannot further interpret the association of altered gambling behavior because many other influencing variables, such as more detailed information on life satisfaction, motivation, substance use, and especially gambling addiction symptomatology, could only be collected at T1: The web-based survey had to be limited to the most important aspects in order to keep the respondent burden low and thus avoid survey fatigue. Therefore, we cannot ensure the validity of the results concerning mental health and life satisfaction required for publication, which is why we have not published the results.

Practical implication
The implications for player protection are explained in more detail. From our point of view, the early detection of problem gamblers is of particular importance. If it turns out that the player is playing beyond his means, he must be excluded. Furthermore, we refer to the need for further research, which we believe is necessary. For example, the influence of advertising on players should be investigated. We have already mentioned in the first version that more research is needed on the risks of online gambling.

We hope that we have made the revisions to your complete satisfaction. We would like to take this opportunity to thank you again for your feedback and for your time.

Reviewer 3 Report

The influence of lockdown on the gambling pattern of Swiss casinos players

Abstract:

  • Line 13 you mention the ‘main characteristics’, please could you add in brackets a couple of examples of these.
  • Line 23 you refer to casinos having player protection measures, as it is online gambling that has increased, do you mean here online casinos?

Method

  • Line 69 - It would be interesting for you to highlight how similar/different rates of gambling is in Switzerland compared to other countries. Are the rates lower, given the blocking measures in place?
  • Line 78 – could you clarify please why players were banned? Was this because they exhibited serious problem gambling behaviours? Could you also clarify what you mean by self-exclusion in the context of gambling?
  • Line 118 – please give a couple of examples in brackets of the different game types. You could include a table with all of the types, and highlight which of those are removed in the Gambling during COVID-19 questionnaire on line 165.

Findings

  • Just a formatting issue (possibly due to the system), I can’t see the whole of the tables, the right hand side is cut off.

Discussion

  • Line 347 – expand on reasons why you think excluded players gamble less during lockdown.
  • Line 400 – sentence not completed.
  • Line 390 – it would be interesting if you could expand upon the implications of your research more. What impact does this have on policy? Especially given the blocking procedures you mention earlier.

Overall, this is a very interesting paper. Thank you.

Author Response

Thank you for the appreciative review. Subsequently you find our response to the comments you have raised.

Abstract:
We agree that the abstract would be improved with the explanations of what exactly is understood by "Main Characteristics". (line 13). Unfortunately, since we have reached the 200-word mark, we can no longer make additions, otherwise we would have to delete other relevant content. However, we take of course your feedback into consideration. In line 78 we specify with some examples what is meant by "main characteristics".

(line 23): (see comment above). The remarks in the abstract refer to land-based gambling.

Method

Prevalence
We discuss the prevalence rate in comparison with other European countries (line 106)

Exclusion
The chapter has been expanded with a section on exclusion (line 109f.)

Types of game
We explain in the flow text which forms of play we included in the survey (line 160-164)

Findings
We clarify with the editor whether there is a need for adjustments to the formatting.

Discussion
 We explain, why the banned players played less intensively during the lockdown. However, we would like to point out that this information should be interpreted with caution due to the partial number of cases.

Line 400
has been corrected

Practical implication

The implications for player protection are explained in more detail. From our point of view, the early detection of problem gamblers is of particular importance. If it turns out that the player is playing beyond his means, he must be excluded. Furthermore, we refer to the need for research, which we believe is necessary. For example, the influence of advertising on players should be investigated. We have already mentioned in the first version that more research is needed on the risks of online gambling.

We hope that we have made the revisions to your complete satisfaction. We would like to take this opportunity to thank you again for your feedback and for your time.